# An Exploratory Analysis of Changes in Circulating Plasma Protein Profiles Following Image-Guided Ablation of Renal Tumours Provides Evidence for Effects on Multiple Biological Processes

**DOI:** 10.3390/cancers13236037

**Published:** 2021-11-30

**Authors:** Tze Min Wah, Jim Zhong, Michelle Wilson, Naveen S. Vasudev, Rosamonde E. Banks

**Affiliations:** 1Division of Diagnostic and Interventional Radiology, Institute of Oncology, St. James’s University Hospital, Beckett Street, Leeds LS9 7TF, UK; jim.zhong@nhs.net; 2Clinical and Biomedical Proteomics Group, Leeds Institute of Medical Research at St James’s, University of Leeds, St. James’s University Hospital, Beckett Street, Leeds LS9 7TF, UK; m.wilson1@leeds.ac.uk (M.W.); N.Vasudev@leeds.ac.uk (N.S.V.)

**Keywords:** image-guided ablation, renal cell carcinoma, radiofrequency ablation, microwave ablation, cryoablation, biomarkers, immune response, danger-associated molecular patterns, proteomics

## Abstract

**Simple Summary:**

Ablation techniques use extremely hot or cold temperatures to kill small cancers. It is now known that in addition to killing the cancer cells, the ablation treatment may stimulate an immune response in patients against the cancer cells, acting like a vaccine. As a result, there is now interest in combining ablation together with drugs that target the immune system of patients with cancer to enhance the effects of both treatments. In order to develop such combined treatments and test them in clinical trials, we need to understand more about their effects so trials can be optimally designed to be most effective. We have analysed 164 circulating proteins in the blood from patients with small renal tumours undergoing ablation treatment to understand more about the effects of ablation on the patient, both at the level of the effects on the cancer cells and the response of the patients.

**Abstract:**

Further biological understanding of the immune and inflammatory responses following ablation is critical to the rational development of combination ablation-immunotherapies. Our pilot exploratory study evaluated the circulating plasma protein profiles after image-guided ablation (IGA) of small renal masses to determine the resultant systemic effects and provide insight into impact both on the tumour and immune system. Patients undergoing cryotherapy (CRYO), radiofrequency ablation (RFA) or microwave ablation (MWA) for small renal tumours were recruited. Blood samples were obtained at four timepoints; two baselines prior to IGA and at 24 h and 1–3 months post-IGA, and a panel of 164 proteins measured. Of 55 patients recruited, 35 underwent ablation (25 CRYO, 8 RFA, 2 MWA) and biomarker measurements. The most marked changes were 24 h post-CRYO, with 29 proteins increasing and 18 decreasing significantly, principally cytokines and proteins involved in regulating inflammation, danger-associated molecular patterns (DAMPs), cell proliferation, hypoxic response, apoptosis and migration. Intra-individual variation was low but inter-individual variation was apparent, for example all patients showed increases in IL-6 (1.7 to 29-fold) but only 50% in CD27. Functional annotation analysis highlighted immune/inflammation and cell proliferation/angiogenesis-related clusters, with interaction networks around IL-6, IL-10, VEGF-A and several chemokines. Increases in IL-8, IL-6, and CCL23 correlated with cryoprobe number (*p* = 0.01, r_s_ = 0.546; *p* = 0.009, r_s_ = 0.5515; *p* = 0.005, r_s_ = 0.5873, respectively). This initial data provide further insights into ablation-induced biological changes of relevance in informing trial design of immunotherapies combined with ablation.

## 1. Introduction

Interventional oncology is now recognised as the fourth pillar of cancer care alongside surgical, medical and clinical oncology [1]. Image-guided ablation (IGA) therapies such as RFA and MWA (heat-based energy), or CRYO (cold-based) are increasingly being used in many cancer types. For patients with renal cancer, this is particularly the case as an alternative to major surgery in the context of patients with small localised renal tumours (<3 cm) or significant co-morbidities, and is included in recent guidelines [2,3]. CRYO is now the predominant modality and outcome data support the effectiveness of IGA for T1a renal tumours [4,5,6].

For patients with metastatic renal cell carcinoma (RCC), immunotherapies such as interleukin-2 have been followed by therapies targeting the vascular endothelial cell growth factor (VEGF) and mTOR signalling pathways. More recently, immunotherapy with immune-checkpoint inhibitors (ICIs) targeting CTLA4 and PD-1 have shown greater efficacy and become first-line treatment, alone or in combination with VEGF receptor-targeted tyrosine kinase inhibitors (TKIs) [7]. However, variable responses and acquired resistance support continued development of new immunotherapeutic strategies [7].

IGA results in immune and inflammatory responses, the nature and extent of which reflects the ablation modality, mechanisms of cell death and injury and tumour type. The release of tumour antigens effectively acts as a “vaccine”, generating anti-tumour responses including occasional “abscopal” effects at sites of disease distant to the primary ablation [8,9,10,11]. Effects are more marked following CRYO, and combination therapies of ablation augmented by ICIs or other immunomodulators are being explored following promising results in preclinical models [8,9,11,12,13]. Proof-of-concept and safety studies show manageable toxicities [14,15] and 35 trials are ongoing or planned (February 2021; clinicaltrials.gov, accessed on 26 November 2021). In renal cancer, small patient-based studies have shown induction of cellular and humoral anti-tumour responses following CRYO plus GM-CSF [16], CRYO-induced clonal expansion of anti-tumour T cells and immune activation in ablated tissue [17] and increases in lymphocytes and Th1 cytokines with CRYO treatment, enhanced by allogeneic NK cells [18]. The Society of Interventional Oncology has highlighted these synergistic opportunities, making recommendations for future directions which include gaining additional understanding of the biological consequences of ablation for each tumour type and ablation modality [19].

Insights into systemic effects of therapies, including TKI-induced Th1 immunity in CML patients [20] and ICI response-associated protein signatures in patients with metastatic melanoma [21], have been obtained using a novel targeted multiplex proteomics profiling platform. With panels focussed on specific disease areas/processes, incorporating both antibodies and PCR-based amplification enhances specificity and sensitivity and allows use with small sample volumes [22]. We hypothesised that using this approach with appropriately selected panels, we would be able to undertake a comprehensive and robust targeted exploration of the systemic changes in patients with small renal masses undergoing IGA to provide new insights and further understanding of the resultant biological changes and highlight relevant factors. The results from this exploratory study will underpin further extended biomarker studies, ultimately contributing to and informing systematic and rational trial design of immunotherapy-ablation combination therapies.

## 2. Materials and Methods

### 2.1. Study Design

Patients were identified and recruited to this prospective, observational study (clinicaltrials.gov, accessed on 26 November 2021, ID: NCT04392076; NIHR CRN CPMS-ID:42803) from the Interventional Oncology clinic at St James’s Institute of Oncology in Leeds, UK between October 2010 and July 2011 (proof-of-concept pilot cohort) and June 2019 to June 2020 (further patients). All participants gave informed written consent (NRES ethics approvals for the Leeds Multidisciplinary Research Tissue Bank (RTB) 10/H1306/7, 15/YH/0080 and 20/YH/0103). Inclusion criteria were: age > 18 years; ability to provide informed consent; undergoing planned image-guided renal tumour ablation with histologically proven malignancy (RCC) or planned biopsy at the time of treatment. Patients all had primary RCC (not recurrent) with a solitary tumour and had not received treatment for RCC previously.

### 2.2. IGA

Patients underwent percutaneous IGA (CT-guided) with CRYO, RFA or MWA as an elective procedure, under general anaesthesia, with hospital discharge usually after overnight observation. All patients routinely received intravenous co-amoxiclav (1.2 g) during the procedure and at 12-h post-IGA. All IGA procedures were performed by one of three consultant interventional radiologists (T.M.W., J.T.S. or J.L.), with experience ranging from 8–17 years. All the renal tumours were treated to achieve a zone of ablation margin with at least a 5 mm margin of normal tissue surrounding the tumour.

RFA was performed with a varying size (3, 3.5 or 4 cm) umbrella shaped multi-tines LeVeen CoAccess RFA needle electrode selected to match the size of the tumour and the pulsed RF current was delivered by an impedance-controlled 200-W generator (Boston Scientific, MA, USA). The RFA needle electrode and the number of overlapping RFA treatments were based on the size and geometry of the tumour. Each RFA treatment protocol was based on the impedance-controlled algorithm of the generator and the treatment effect was achieved when the ‘impedance roll off’ where coagulation necrosis had been reached which was followed by concurrent power shut down of the generator. For MWA, tumours were targeted with the single straight tip Neuwave PR electrode and microwave was delivered by the MWA generator using the protocol of 65W over 5 min (Ethicon, J&J, NJ, USA) using the protocol of 65 W for 5 min for the two renal tumours treated. For CRYO, between 4–11 cryoprobes were inserted depending on the size and geometry of the renal tumours, and all renal tumours were treated with the protocol of two freeze-thaw treatment cycles—10 min freezing following by 4 min passive thawing and 1 min of active thawing. The ice-ball was formed at the tip of the cryoprobes to cover the whole tumour using the Joule-Thomson effect with argon-helium gas delivery from the CRYO generator (Boston Scientific, MA, USA).

### 2.3. Blood Sample Collection and Processing

Venous blood samples for plasma (1 × 4 mL Greiner EDTA tube) were collected from each patient at four timepoints (TP). The scheduled TPs comprised at least one sample, preferably two, prior to ablation; the inclusion of two baseline samples wherever possible was to allow the intra-individual variability in each protein to be assessed, against which changes following ablation could be compared. Specifically, the first baseline sample (TP1) was collected at up to 3 months prior to ablation, a reference baseline sample (TP2) within 24 h prior to ablation, a sample (TP3) at 24 h post-ablation, and a final sample (TP4) at 1–3 months post-ablation. All samples were processed according to standardised operating procedures. Between 45 min and up to a maximum of 2 h post-venepuncture, samples were centrifuged for 10 min at 2000× *g*, 4 °C, and the plasma removed, aliquoted and stored frozen at −80 °C. In addition, plasma samples from 11 healthy controls processed and stored in the same way were accessed from the Leeds Multidisciplinary RTB.

### 2.4. Proteomic Analysis of Circulating Plasma Proteins

EDTA plasma samples were analysed on Olink Target Oncology II and Immuno-Oncology panels using the Fluidigm Biomark platform, together with internal QC samples (www.olink.com/products/, accessed on 26 November 2021). The initial pilot cohort was analysed in November 2016 and the remaining samples in July 2020. Samples from each patient were batched on the same plate runs but within each plate, all patient and control samples were randomised. To allow direct comparability of the results over time, 10 samples analysed in 2016 were also reanalysed in 2020 (“bridging controls”).

### 2.5. Statistical and Bioinformatic Analysis

Data were exported in Excel with results recorded as NPX units, a composite value based on the qPCR output and the normalisation routinely performed by OLink to minimise intra-and inter-assay variation (www.olink.com/application-category/white-papers/, accessed on 26 November 2021). Data were further normalised to account for differences across analysis period using the bridging controls (www.olink.com/question/how-can-i-compare-results-from-two-different-studies/, accessed on 26 November 2021). As quantification is relative, results can be compared across samples for any protein but different proteins with the same NPX values may differ in concentration.

After converting NPX values from log2 to a linear scale, statistical analysis was undertaken within Graphpad Prism with matched-pair analysis across the different timepoints undertaken for each protein using the Wilcoxon signed rank test (significance criteria of *p* < 0.05 and false discovery rate (FDR) *q* < 0.01 to correct for multiple testing). Associations of clinical parameters with post-IGA changes in proteins and correlations between proteins were assessed using Spearman’s rank correlation. Using R (MW), percent change plots, using the results from the baseline sample 24 h pre-ablation (TP2) as a reference, were created with ‘ggplot2’ [23], and heatmaps with ‘ComplexHeatmap’ [24]. To assess the intraindividual variability for each protein, the values of the TP1 sample for each protein were expressed as a percentage of TP2 for each patient and the median and inter-quartile ranges determined.

Bioinformatic analysis of the functional classification of any proteins changing significantly was undertaken using DAVID (v6.8; www.david.ncifcrfgov/, accessed on 9 February 2021) [25] using the default settings and known and predicted associations (direct and functional) were examined using the STRING database (v11.0; www.string-db.org, accessed on 23 February 2021) [26].

## 3. Results

### 3.1. Patient Characteristics and Treatment

Overall, 55 patients were recruited, 35 of whom underwent ablation and with suitable sample collection within the study timeframe (Figure 1, Table 1 and Appendix A). Median tumour size was 3.0 cm (range 1.9–5.4 cm) and the majority (71%) were clear cell RCC. Overall, 25/35 patients (71%) were treated with CRYO although across the two periods of recruitment, the predominant treatment modality changed from RFA to CRYO. Three patients experienced acute complications post-ablation and there were no major differences in the range of comorbidities across the cohorts (Appendix A). Sample collection was largely complete, with all patients providing at least one baseline sample and 27/35 patients providing two baseline samples. For three patients, a final long-term (TP4) post-ablation sample was unavailable. The healthy control cohort consisted of 11 volunteers, 5 male and 6 female with an age range of 51 to 73 (median 63).

### 3.2. Baseline Intra-Individual Variability of Plasma Proteins and Comparison with Healthy Controls

In total, 164 different proteins were measured in each sample (92/panel but 20 overlapping across panels), 6 of which were below the lower level of quantitation in the majority of samples (IL-33, IL-1alpha, IL-2, CD28, IL-13, IL-4). Evaluating the normal intra-individual variability of each of the measured proteins over time between the two baseline samples (TP1 and TP2) for each patient allowed the significance of any changes seen post-ablation to be interpreted. The median time difference between TP1 and TP2 was 33 days (2–231 days). No proteins changed significantly between TP1 vs. TP2. Comparison of baseline samples with those from healthy controls showed 16 proteins to be present at significantly higher concentrations in the RCC patients (Appendix A) although there was considerable overlap between groups.

### 3.3. Plasma Protein Changes Following CRYO

Focussing initially on the CRYO-treated group and comparing the baseline reference TP2 (for two patients this was not available and TP1 was used) with TP3, 46 proteins differed significantly with 29 increasing and 17 decreasing post-CRYO (Table 2, Table 3, Appendix A, Figure 2 and Figure 3, respectively, as examples; all proteins—Appendix A). Of these, 26 were on the immuno-oncology panel and 14 on the oncology II panel with a further 6 common to both. When the proteins were ranked in terms of numbers of patients showing the TP3 post-ablation increase (Table 2), there was marked heterogeneity ranging from 100% of patients showing changes in IL-6 (although varying from 1.7 to 29-fold increase in concentration) and 92% in IL-10, compared with only 50% in CD27, VEGF-A and TNFSF13, when baseline variability was taken into account. Similarly, when the patients were ranked in terms of numbers of proteins increasing significantly, two patients had increases in 29/29 significant proteins, whereas at the other extreme, one patient showed an increase in only 11/29 proteins. A heatmap of the data following unsupervised clustering showed evidence of a hierarchical pattern to the changes (Figure 4a). Three major protein clusters can be seen with some “core” proteins (the left most cluster) changing in most patients and additional proteins from either or both of the other two clusters being sequentially superimposed on these (i.e., moving from left to right in Figure 4a) in different patient clusters, reflecting greater systemic response. In the patients in the least reactive cluster, only the “core” proteins changed in most cases. A similar trend was seen in the proteins which decreased (i.e., moving from left to right in Figure 4b). Although patients 14 and 21 showed high numbers of both increasing and decreasing proteins, for most patients this was not the case and overall, there was a significant negative correlation between the numbers of increased and decreased proteins per patient (*p* = 0.0152, Spearman r = −0.4895). Interestingly histological subtype of RCC did not seem to be a key driver of protein patterns (Figure 4 and Appendix A) although this would need to be examined in much larger studies with additional non-clear cell cases to allow formal statistical comparison. Three of the patients who were treated with CRYO developed acute complications (Appendix A). When the longitudinal profiles were examined for this treatment group were examined with these patients highlighted (Appendix A), the patient who developed fever appeared to show the largest response in several inflammatory proteins in particular and was one of the two patients described above with increases in all 29/29 significantly increasing proteins post-CRYO. Examining associations with further clinical parameters showed the magnitude of the CRYO-induced increases in three proteins (IL-8, IL-6, and CCL23) to be significantly positively correlated with the number of cryoprobes, and with three proteins (pleiotrophin, RSPO3 and TNFSF13), being negatively correlated (Figure 5).

Comparing the reference baseline pre-ablation TP2 with the longer-term post-ablation samples (TP4), seven proteins differed significantly and had not returned to pre-treatment levels. Four of these (MMP12, RSPO3, WFDC2 and CD27) were amongst those also showing significant changes 24 h post-CRYO (TP3) and with three (TNFSRF19, PVRL4 and TGFR2) not reaching significance until TP4. The most striking of these was MMP12 (Figure 3) which decreased significantly at TP3 but showed a rebound increase at TP4. The median time difference between ablation and TP4 was 49 days (range 6–188 days) but all except one were at least 27 days. Interestingly, although not reaching statistical significance in the group as a whole, seven patients showed coordinate increases at TP4 in several proteins involved in the immune response including CD40L, IL-7, CD4, PD-L1, IFN-gamma and CD83 (Appendix A).

### 3.4. Plasma Protein Changes Following RFA or MWA

In contrast, no proteins changed significantly following RFA (*n* = 8). Similar trends were seen as for CRYO but generally less marked (Figure 2 and Figure 3 and Appendix A). With only two patients receiving MWA, statistical analysis was not undertaken but again changes were much less marked.

### 3.5. Functional Characteristics of Significantly Changing Proteins Post-CRYO

When the functional annotation clustering was examined for the proteins that were significantly changed post-CRYO, clusters included those involved in immune and inflammatory/cytokine response, cell proliferation, regulation of apoptosis and regulation of angiogenesis including response to hypoxia and white cell chemotaxis. The main GO biological functions associated with the two largest clusters, namely immune/inflammation-related and cell proliferation/angiogenesis-related, each of which contained 30 overlapping proteins, are shown in Figure 6a. When the predicted/known interactions, both direct physical and functional association were mapped using STRING, the majority of significantly changing proteins clustered within these networks with particular focusses around IL-6, IL-10, VEGF-A and several chemokines (Figure 6b). Several of the proteins were significantly correlated with each other (Appendix A) including IL-10 and IL-15 (*p* < 0.001, rs = 0.8922), angiopoietin-1 and PDGF-B (*p* < 0.001, rs = 0.853), HGF and ESM-1 (*p* < 0.001, rs = 0.8121) and TFPI-2 and ESM-1 (*p* < 0.001, rs = 0.8087), largely aligning with the proposed clustering (Figure 4a) and functional associations (Figure 6b).

## 4. Discussion

Although surgery is the mainstay of treatment for patients with localised renal cancer and with ablation increasingly used for smaller tumours, treatment of metastatic RCC and the development of adjuvant and neo-adjuvant treatments has undergone a revolution in the last three decades. Resistant to conventional chemotherapy and radiotherapy treatment, increasing knowledge of the natural history of this cancer has underpinned the development of treatment strategies. Reports of spontaneous regression, particularly following nephrectomy, the demonstration of tumour-associated antigens and tumour-specific T-cells and tumour-associated immunosuppression led to initial immunotherapy treatments with cytokines such as IL-2 [27]. As the underlying genetic drivers involved in RCC development were determined with their impacts on biological pathways, therapies targeting the vascular endothelial cell growth factor (VEGF) and mTOR signalling pathways were developed [7,28,29]. Interestingly although genetic analyses have shown renal cancer to have a relatively low number of driver mutations arising from single nucleotide variants, a recent study has highlighted this cancer type as having a high proportion of insertion and deletion mutations which may generate neoantigens and account for its immunogenic phenotype [30]. With the development of ICIs targeting CTLA-4 and PD-1 to enhance antitumour T cell activity, these immunotherapies are the current first-line treatments [7,28,29]. Based on a knowledge of the relative impacts of the treatments on both the cancer itself and the tumour microenvironment, combination therapies are also now being explored involving single-agent TKIs and ICIs [7,29] and more recently a pilot study of anti-CTLA-4 (tremelimumab) in combination with CRYO, which showed a significant increase in CD3+ and CD8+ T cell infiltration) in the tumour microenvironment compared with tremelimumab monotherapy [31]. Interestingly, these effects were restricted to the clear cell RCC histological subtype. This was hypothesised to be due to the necrotic cell death caused by ablation and the release of intracellular danger signals and activation of antigen-presenting cells. The balance of apoptosis versus necrosis is likely to be critical in determining the immunostimulatory or immunosuppressive impacts of the treatment, and in our study we have described evidence for changes in both elements following CRYO, in terms of proteins involved in apoptotic regulation, and, for the first time, DAMPs.

Trials bridging interventional oncology and immunotherapy need to be designed systematically, with important balances to be achieved between the recognised unwanted pro-tumourigenic effects of the ablation consequent to the inflammatory response, and the immunostimulatory effects needed to promote anti-tumour responses [9,19]. Our study provides the most comprehensive profiling yet of the biological changes post-IGA, which are detectable systemically without the need for tissue biopsy. Impacts on biological processes including inflammation, the immune response, cellular damage and repair through to hypoxia-mediated events and vascular effects are apparent. These are likely to reflect an integrated effect on the tumour itself and the host response, with proteins released by damaged or dying cells or secreted or shed as part of a coordinated response. However, to what extent any changes seen are due to direct effects of the ablation on the tumour cells or microenvironment as opposed to an indirect effect arising for example from ablation of tumour cells with consequent removal of local tumour-associated immunosuppressive factors is not clear. Importantly, by focussing on a single tumour tissue, i.e., the kidney, we demonstrate the considerable inter-individual variation in the magnitude and type of response even within such a group, in addition to the modality-dependent effects seen. This will be important in-patient stratification [19].

Effects of ablation on tumours are highly dependent on whether hyperthermic causing protein denaturation or freezing, which generates a more marked immune response potentially through preserving antigenicity [8,9,13]. We found CRYO to produce the most marked effects on circulating proteins. In the centre of the freeze zone, cell death occurs as a direct result of physical damage due to ice formation and osmotic effects, and from hypoxia due to vascular damage, generating a necrotic core post-thaw. Across the temperature gradation from −50 °C centrally through the transition zone and to 0 °C in the periphery, diverse additional effects include cell injury, which may be reversible, and activation of cell stress pathways leading to apoptosis, vascular stasis and thrombosis and generation of immune and inflammatory mediators [8,10,13]. We show ablation-induced changes in many proteins involved in these processes, either pro-inflammatory and chemotactic such as IL-6, IL-8, IL-15 and CCL23, recruiting macrophages, neutrophils, NK cells, T cells and dendritic cells as part of the innate immune response, proteins reflecting vascular involvement and angiogenesis such as ESM-1, VEGF, HGF and PGF, or hypoxia-responsive including CA-IX, VEGF, PGF and HO-1. Antigens released during necrosis together with the inflammatory response and antigen presentation can stimulate cytotoxic T-cell development within the adaptive immune response. However, under certain conditions, “immunogenic cell death” rather than anergy also results from stress-induced regulated cell death processes such as apoptosis and necroptosis under certain conditions [32]. Endogenous danger signals such as histones, mitochondrial DNA, HMGB1 and HSP70 arising from cell death, damage or stress are critical (“damage-associated molecular patterns”; DAMPs), activating macrophages via routes including phagocytosis or interaction with toll-like receptors [32,33]. Importantly as well as detecting changes in proteins implicated in regulating apoptosis such as TRAIL, FASL, CD27 and galectin-1, we show for the first time, ablation-induced changes in circulating DAMPs including decorin, syndecan and S100 proteins [33]. Ultimately, the balance of immunostimulatory and immunosuppressive factors and positive and negative regulatory controls in the tumour microenvironment will determine if anti-tumour responses are generated, influenced by tumour-derived factors including mechanisms of cell death and antigen release and the individual patient’s immune system [8,9,13].

The pro-oncogenic activity of many of the proteins induced following ablation [19] is exemplified here by IL-6, VEGF and HGF. Although early post-ablation, cytokines such as IL-6 and IL10 dominated, a coordinate increase in some Th1 cytokines such as IFN-gamma and CD83 as a marker of activated dendritic cells was apparent later, but only in a subgroup. Clearly kinetics are important and results are timepoint-dependent snapshots. In a murine breast cancer model, a novel cryo-thermal treatment produced an IL-6 driven acute phase protein cluster predominating in the first 48 h, followed in subsequent weeks by tumour progression-associated proteins [34]. Analysis of splenic tissue at day 3 provided some evidence for a Th2 to Th1 shift with increased IFN--gamma and IL-12 [34]. Following MWA of hepatic tumours, changes at 24 h were limited to a small increase in IL-2 but far greater changes were apparent 16 days later including in IL-6, IL-10, IL-8 and VEGF, with IL-6 being weakly correlated with ablation energy [35]. In patients with renal cancer, next-generation expression analysis has shown CRYO-induced increased tissue infiltration by antigen-presenting cells and increased oligoclonal expansion of T cells in 15/22 patients three months-post treatment, some of which were also detected peripherally [17].

Comparison of the impacts of ablation modality on systemic changes include induction of far greater increases in IL-6 and IL-10 post-CRYO in a rat liver model (and AST/ALT implying greater cell leakage) compared with RFA, despite comparable levels of tissue necrosis [36]. Our reported increases in IL-6 and IL-10 post-IGA are similar to previous reports in several tumour types [37] and similarly the most marked changes were seen following CRYO in this earlier study although this was based on only 4 patients treated with CRYO compared with 31 treated with RFA or MWA [37]. Changes were also concluded to be tumour tissue type-dependent but with limited numbers (*n* = 1 to 4 for most different tumour types) this is preliminary, particularly given the marked inter-individual variability we describe even within one tumour tissue type. Such inter-patient variability was also described in breast cancer patients treated pre-operatively with CRYO, ipilimumab or combination in terms of the changes in cytokines including IL-8 and IFN-gamma in particular, although the extent to which the surgery also contributed is not clear [38]. From inspection of the protein profiles following CRYO, the remarkable consistency of changes in relation to the CRYO timing for some proteins across patients and the relationship of some proteins with cryoprobe number supports ablation-dependent effects rather than random changes. However, contributions to inter-patient variability in terms of the magnitude of the changes in proteins post-ablation may also arise from acute clinical complications which may confound data interpretation. For example, a patient with fever post-ablation showed the largest changes in several of the inflammatory proteins and so the changes seen likely reflect both cryoablation and infection in this case.

Consideration of comorbidities in biomarker measurements is also important, both in terms of impact on the baseline starting point of any analyte and the interpretation of treatment-induced changes but also in terms of possible effect on the response of the patient to the ablation. When comparing the baseline values of the patients with results obtained from healthy controls of similar age, with the caveat of the small size of the group, 16 proteins were significantly increased in the patient group. These may be directly or indirectly tumour-related such as IL-6 which is known to be increased in patients with RCC, although these are only small renal masses. However, in a large-scale study examining the impact of genetics and lifestyle of protein profiles, several of these have been identified as being associated with other diseases, for example amphiregulin is increased in diabetes, myocardial infarction and stroke and indeed increased IL-6 has also been associated with stroke and diabetes [39]. Clearly intrinsic patient variability will be important to consider together with clinical and technical aspects. Such complexities are illustrated by our findings of cryoprobe number-dependent changes in circulating proteins which we interpret as reflecting the greater inflammatory response from the larger ablation zone with increasing number of cryoprobes. A previous study has demonstrated freezing rate-dependent changes in numbers of specific T cells and outcome in a mouse breast cancer model [40], hypothesised to be due to differences in the balance of necrotic versus apoptotic cell death. From a clinical perspective, such protein or cellular readouts may help in the standardization of treatment approaches and the assessment of the magnitude and duration of the ablation-induced biological response, which will be important in optimizing combination therapy strategies.

Our study is exploratory and observational and although a strength is its focus on one tissue type, the inherent heterogeneity within this group and overall sample size of 36 patients limits some aspects. For example, with almost 70% of cases being of clear cell subtype, even in the largest treatment group (CRYO) there are only 4 patients with papillary RCC compared with 16 clear cell RCC making comparisons difficult. Similarly, the small number of patients treated with MWA in particular, limits comparisons between ablation modalities. Larger sample sizes would also allow any potential confounding influences to be explored such as potential effects of medication. In our study all patients underwent similar general anaesthetic and antibiotic regimens initially and so these are unlikely to have influenced comparisons across the groups. Most studies analyse samples which have been taken after an operative procedure and so the impact of anaesthesia alone is not clear. A recent study involving 59 healthy volunteers found that at 5 h after induction of anaesthesia there were no or insignificant changes in plasma IL-6, TNF-alpha or CRP [41]. In contrast a much larger study profiling 983 proteins in patients with ovarian cancer found that a large number of these proteins were significantly different between an awake cohort at diagnosis compared with a separate cohort who had fasted and undergone anaesthesia for 10–45 min [42]. However, this study is limited in that different patients were included in the different arms of the study with varying cancer stage mixes, differences in sample collection and processing and potentially differences in co-morbidities and similar factors, which may have confounded the interpretation of these results. Even with the current sample size the influence of cryoprobe number on several of the proteins was identified in our study, which is important. Future studies would also benefit from including a larger number of timepoints so that the kinetics of the various protein changes could be analysed in more detail. An additional limitation of the current study is the variability of timing of the final sample timepoint (TP4) due to its dependency on the hospital attendance of the patient for clinical purposes, and which would benefit from more rigorous standardisation, which would be possible if managed in the context of a clinical trial. Such a trial scenario should also overcome the single site limitation of the current study meaning results would be more generalisable, although inevitably by their very nature, the majority of initial exploratory studies such as this would be expected to be carried out on a single site basis.

As future clinical trials are developed examining ablation alone and in combination with systemic treatments such as the ICIs, it will be important to incorporate similar biomarker profiling studies in order to systematically optimise trial design by understanding the biological changes [19]. Our study has added to the existing literature in several aspects, providing information that will help to develop further such studies. The potential of the proteomic platform used has enabled the most comprehensive profiling to date, which has allowed confirmation of some existing findings, in particular the inflammatory response and the more marked response following CRYO. However, importantly it has enabled novel findings such as ablation-induced changes in circulating DAMPs and a more extensive description of the balance of the immunostimulatory and immunosuppressive factors detectable systemically without the need for tissue biopsy. Future studies may benefit from exploring additional protein panels to widen the information base, and investigate whether immunophenotyping of peripheral blood cells provides complementary information. Additional consideration should also be given to parallel exploration of lipidomics/metabolomics. The relevance to the development of immunotherapy-ablation combination therapies is immediately apparent with the importance of lipid moieties being increasingly recognised, both in terms of their dysregulation in RCC, which provides both diagnostic and therapeutic targeting opportunities [43,44], and in their major roles in the regulation of immune and inflammatory responses [45,46]. In terms of future study design, our data show the heterogeneity of the response at an individual patient level, even within a histological subtype, and this will need to be factored in to determine sample size, the importance of including two baseline pre-treatment samples to determine the normal intra-patient variability and derive appropriate cut-offs that are protein-specific, and illustrated potential confounding factors that will need to be explored and optimised based on the modality selected, such as cryoprobe number. Ultimately further exploration of these factors will guide optimal scheduling of the combination therapies and may identify biomarkers that would be useful as predictive biomarkers, supporting patient stratification and monitoring of response with associated benefits in terms of optimising treatment, minimising exposure to toxicity and maximising health economic benefits.

## 5. Conclusions

With the increasing interest in developing therapeutic strategies for several cancer types involving combining ablation and immunotherapy, it is essential that clinical trials are designed rationally, based both on existing biological knowledge but also through including translational elements to further strengthen the knowledge base. Through evaluating circulating plasma profiles in patients after image-guided ablation (IGA) of small renal masses (RCC) using a multiplex proteomic platform, our pilot exploratory study has demonstrated systemic changes in proteins involved in several biological processes. These were principally in regulating inflammation, danger-associated molecular patterns (DAMPs), cell proliferation, hypoxic response, apoptosis and migration, and likely reflect both effects on the tumour and release of tumour-derived proteins as well as systemic responses.

Significantly, several factors were identified as being particularly important in determining the protein response seen. Ablation modality was critical with the most pronounced changes being seen following CRYO, although this needs exploring further, with larger numbers of patients receiving MWA in particular. Within CRYO, the number of cryoprobes used correlated significantly with several of the protein changes. Although intra-individual variation was low for most proteins over time, some proteins were much more variable, which is important in determining which changes were significant following ablation. However, there was also marked heterogeneity between patients as to the nature and extent of response seen in circulating proteins and it will be important to understand this further in terms of potential relevance to response to treatment and the importance of other factors such as co-morbidities and post-ablation complications. These results provide additional biological insights into the effects of ablation, highlight the potential of the specific multiplex proteomic platform in deriving such information and some of the factors influencing the ablation-related effects, and illustrate some of the critical factors to explore further in designing future clinical trials and in developing predictive or monitoring biomarkers.

## Figures and Tables

**Figure 1 cancers-13-06037-f001:**
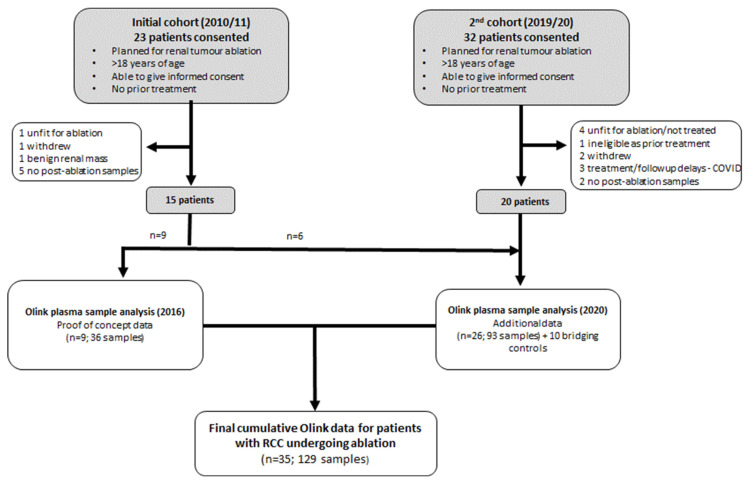
Flow diagram summarising the recruitment of patients to the study.

**Figure 2 cancers-13-06037-f002:**
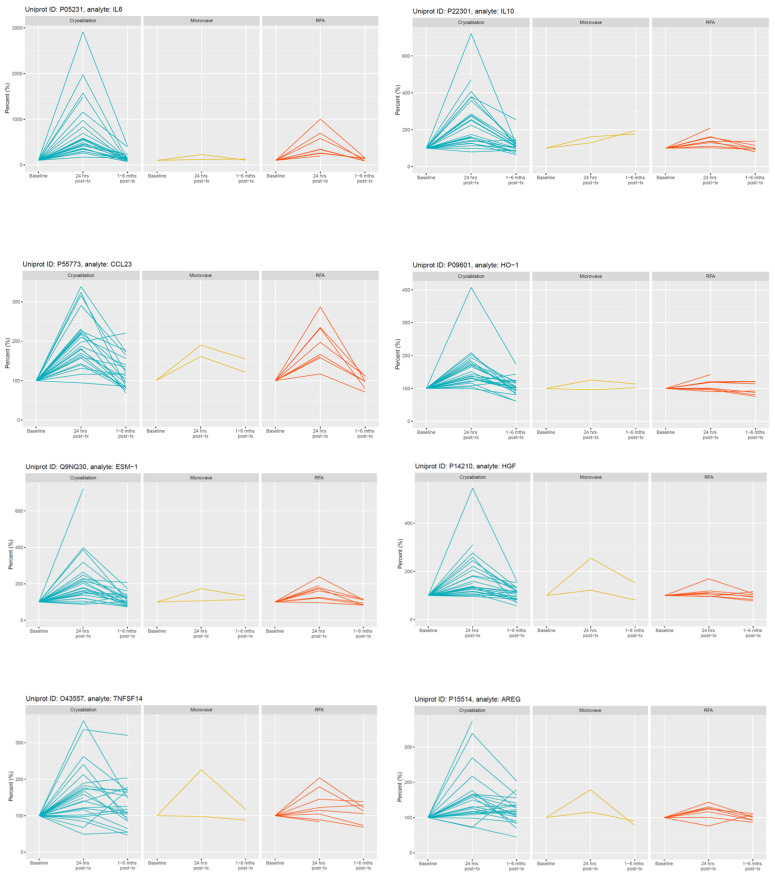
Selected examples of longitudinal changes in plasma proteins which increased significantly at 24 h post-CRYO ablation. Results for each protein are expressed as a % of the reference baseline TP2; (100%) and are shown for patients receiving CRYO (*n* = 25), MWA (*n* = 2) and RFA (*n* = 8). Note the y-axis scale is optimised for each protein given the variation in magnitude of changes.

**Figure 3 cancers-13-06037-f003:**
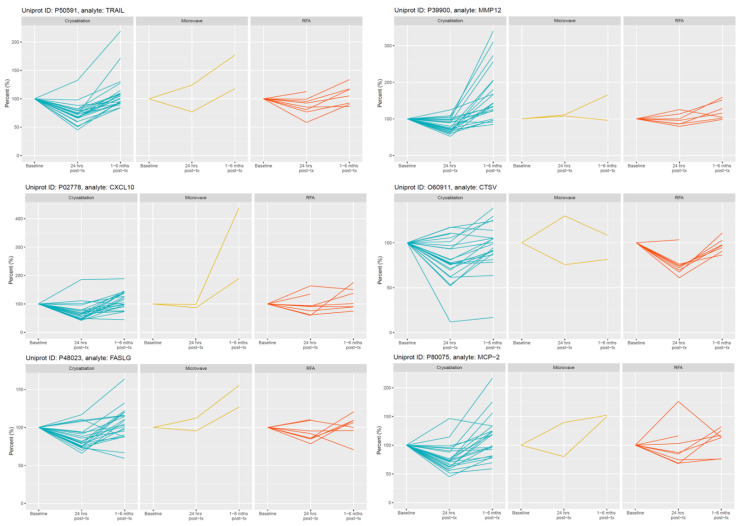
Selected examples of longitudinal changes in plasma proteins which decreased significantly at 24 h post-CRYO ablation. Results for each protein are expressed as a % of the reference baseline TP2; (100%) and are shown for patients receiving CRYO (*n* = 25), MWA (*n* = 2) and RFA (*n* = 8). Note the y-axis scale is optimised for each protein given the variation in magnitude of changes.

**Figure 4 cancers-13-06037-f004:**
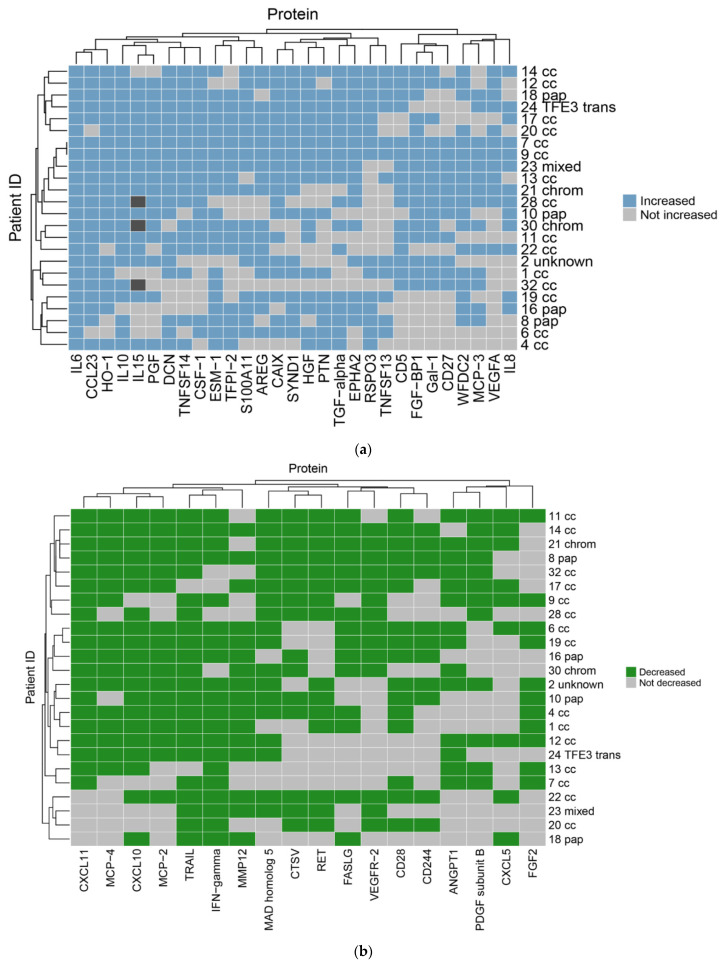
Heatmaps for proteins (**a**) increasing and (**b**) decreasing significantly post-CRYO. The readout is binary with each patient scored based on being above or below the threshold cut-point for each protein after taking into account background intra-individual variability. The blue and green squares indicate the protein has met the threshold of an increase (**a**) or decrease (**b**), respectively, for each patient individually, with light grey indicating it has not. Dark grey indicates missing data for 3 patients for one protein. The clustering of the patients is indicated at the left-hand side with the corresponding individual patient identifiers at the right-hand side together with the histological subtype (cc, clear cell; pap, papillary; chrom, chromophobe; TFE3 trans, TFE3 translocation). The clustering of the proteins is indicated above each plot with the corresponding protein identifiers below each plot.

**Figure 5 cancers-13-06037-f005:**
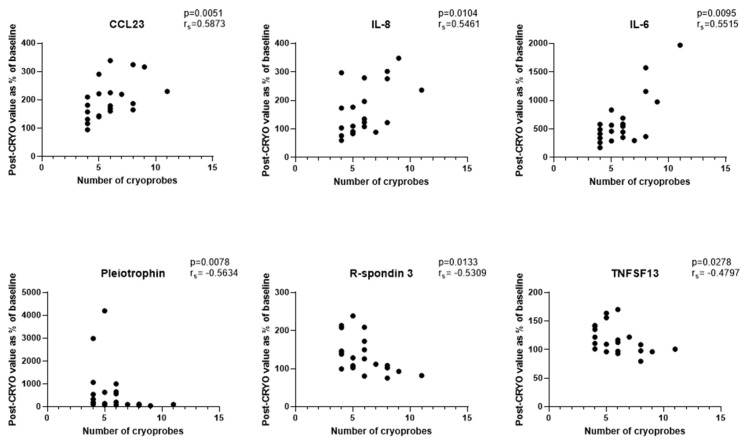
Correlations between numbers of cryoprobes and the magnitude of changes in plasma proteins 24 h post-CRYO (*n* = 21 patients as 3 patients with acute complications post-ablation and 1 patient whose 24 h sample failed analytical QC excluded).

**Figure 6 cancers-13-06037-f006:**
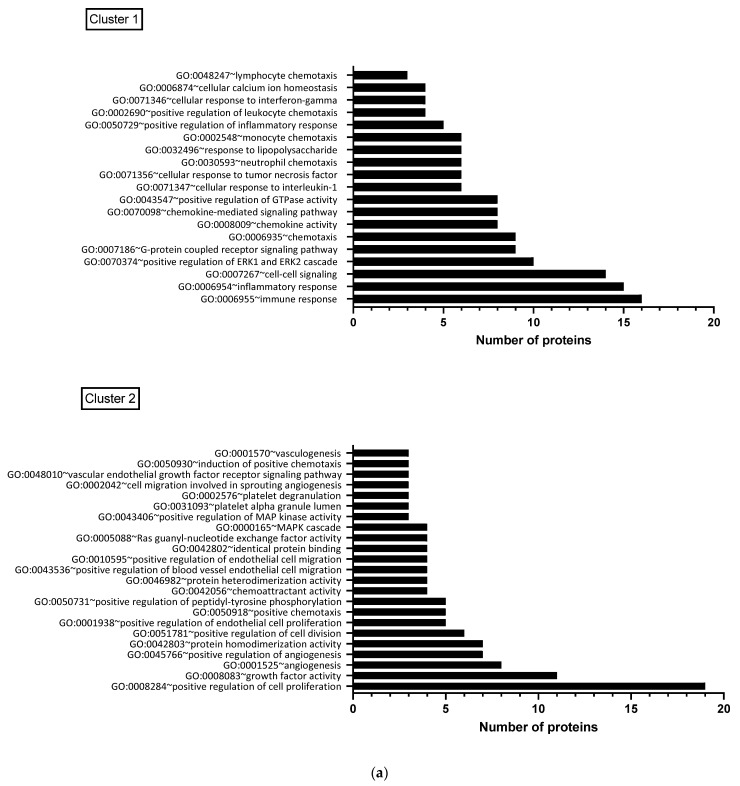
(**a**) Functional clustering of proteins which changed significantly post-CRYO, based on gene ontology terms. Results are shown for the two largest clusters, each of which contained 30 overlapping proteins, with the number of proteins associated with each GO biological function shown: Cluster 1 was predominantly immune/inflammation-related and Cluster 2 was focussed on cell proliferation and angiogenesis. (**b**) Predicted and known protein interactions of the proteins highlighted as significantly increasing or decreasing post-CRYO based on the STRING database (26). This includes both predicted and known and is based on both functional associations and direct interactions. Known interactions: turquoise line (curated databases) and pink line (from experimental data). Predicted interactions: bright green line (gene neighbourhood). Others: Light green line (text mining), black line (co-expression) and lilac line (protein homology).

**Table 1 cancers-13-06037-t001:** Patient demographics and tumour characteristics (*n* = 35).

Parameter	Value
**Age** median (range)	70 (38–86)
**Sex**	Male (22) Female (13)
**Treatment Type**	
CRYO	25 (71.4%)
RFA	8 (22.9%)
MWA	2 (5.7%)
**Tumour Size (cm);** median (range)	
**Histopathology**	
Clear cell RCC	25 (71.4%)
Papillary RCC	4 (11.4%)
Chromophobe RCC	3 (8.5%)
Xp11.2/TFE3 translocation RCC	1 (2.9%)
Unclassified RCC	1 (2.9%)
Insufficient sampling	1 (2.9%)

**Table 2 cancers-13-06037-t002:** Plasma proteins increasing significantly 24 h post-CRYO (TP3) compared with the reference baseline assessment (TP2) in the 24 h prior to CRYO (*n* = 24 as one TP3 sample failed analytical QC) in a matched pair analysis (Wilcoxon signed rank test). Results are presented as percentage change for TP3 compared with baseline, with linear NPX protein results presented in Appendix A. The number of patients showing an increase of any magnitude in each protein is presented together with the number that also exceeded the baseline protein-specific variability. * Due to changes in the Immuno-oncology panel, measurements for protein P40933 were only available in the 2020 dataset and so *n* = 20 for that protein. ** Four proteins were on both panels with similar results for both, with those from the Immuno-oncology panel below.

Protein	Uniprot Accession Number	TP3 Protein Result as % of TP2 Baseline; Median (Range)	Number of Patients with Increase	Number of Patients with Increase > IQR of Baseline Variability	*p* Value	*q* Value (FDR)
Interleukin-6 (IL-6)	P05231 **	518.7 (172.1–2917)	24 (100%)	24 (100%)	<0.001	<0.001
Interleukin-10 (IL-10)	P22301	199.6 (79.6–721)	22 (91.7%)	22 (91.7%)	<0.001	<0.001
C-C motif chemokine 23 (CCL23)	P55773	192.3 (94.4–338.6)	23 (95.8%)	22 (91.7%)	<0.001	<0.001
Heme oxygenase 1 (HO-1)	P09601	140.9 (99.6–407.1)	23 (95.8%)	21 (87.5%)	<0.001	<0.001
Endothelial cell-specific molecule 1 (ESM-1)	Q9NQ30	177.7 (87.1–720.9)	22 (91.7%)	21 (87.5%)	<0.001	<0.001
Decorin (DCN)	P07585	116.8 (86.1–175.1)	20 (83.3%)	20 (83.3%)	<0.001	<0.001
Interleukin-15 (IL-15)	P40933 *	154.2 (77.3–229.1)	16 (80%)	16 (80%)	<0.001	<0.001
Carbonic anhydrase-IX (CA-IX)	Q16790 **	147.6 (76.9–473.2)	21 (87.5%)	19 (79.2%)	<0.001	<0.001
Macrophage colony stimulating factor 1 (CSF-1/M-CSF)	P09603	116.8 (99.4–237.5)	22 (91.7%)	18 (75%)	<0.001	<0.001
Tumour necrosis factor ligand superfamily member 14 (TNFSF14)	O43557	154.1 (49.1–360.4)	19 (79.2%)	18 (75%)	<0.001	0.002
Syndecan-1 (SYND1)	P18827	174.3 (82.2–515.6)	22 (91.7%)	18 (75%)	<0.001	<0.001
Transforming growth factor-alpha (TGFA)	P01135	129.7 (81.2–488)	19 (79.2%)	18 (75%)	<0.001	0.002
Hepatocyte growth factor (HGF)	P14210 **	145.0 (95.9–545.7)	23 (95.8%)	18 (75%)	<0.001	<0.001
WAP-four disulfide core domain protein 2 (WFDC2)	Q14508	130.4 (64.3–197)	22 (91.7%)	18 (75%)	<0.001	<0.001
Amphiregulin (AREG)	P15514	140.5 (71.7–373.1)	21 (87.5%)	18 (75%)	<0.001	0.002
Protein S100-A11 (S100-A11)	P31949	126.5 (79.6–224.5)	20 (83.3%)	18 (75%)	<0.001	<0.001
Ephrin type-A receptor 2 (EPHA2)	P29317	114.1 (58.6–246.7)	20 (83.3%)	17 (70.8%)	<0.001	0.004
Placenta growth factor (PlGF)	P49763	126.0 (79.6–235.7)	20 (83.3%)	17 (70.8%)	<0.001	0.001
Fibroblast growth factor-binding protein 1 (FGFBP1)	Q14512	125.7 (76.1–706.6)	18 (75%)	17 (70.8%)	0.001	0.005
T cell surface glycoprotein CD5 (CD5)	P06127	118.8 (84.7–192.5)	20 (83.3%)	16 (66.7%)	<0.001	<0.001
Pleiotrophin (PTN)	P21246	207.2 (49.8–4195.3)	19 (79.2%)	16 (66.7%)	<0.001	0.002
Tissue factor pathway inhibitor 2 (TFPI-2)	P48307	134.7 (47.5–290.1)	19 (79.2%)	16 (66.7%)	<0.001	0.004
Galectin-1 (GAL-1)	P09382	111.8 (74.5–183.6)	18 (75%)	15 (62.5%)	0.002	0.003
R-spondin 3 (RSPO3)	Q9BXY4	133.3 (75.3–477.4)	19 (79.2%)	15 (62.5%)	<0.001	0.002
Monocyte chemotactic protein 3 (MCP-3)	P80098	152.6 (52.9–330.5)	19 (79.2%)	14 (58.3%)	<0.001	0.001
Interleukin-8 (IL-8)	P10145	154.1 (59.7–348.1)	18 (75%)	13 (54.2%)	<0.001	0.002
Vascular endothelial growth factor-A (VEGF-A)	P15692	124.3 (80.6–230.6)	18 (75%)	12 (50%)	<0.001	0.002
CD27 antigen (CD27)	P26842 **	110.5 (85.5–170.1)	21 (87.5%)	12 (50%)	<0.001	<0.001
Tumour necrosis factor ligand superfamily member 13 (TNFSF13)	O75888	112.5 (71.8–245.1)	18 (75%)	12 (50%)	0.001	0.005

**Table 3 cancers-13-06037-t003:** List of proteins decreasing significantly 24 h post-CRYO compared with the baseline assessment in the 24 h prior to CRYO (*n* = 24 as one TP3 sample failed analytical QC) in a matched pair analysis (Wilcoxon signed rank test). Results are presented as percentage change for TP3 compared with baseline, with linear NPX protein results presented in Appendix A. The number of patients showing a decrease of any magnitude in each protein is presented together with the number that also exceeded the baseline protein-specific variability. * Two proteins were on both panels with similar results for both, with those from the Immuno-oncology panel below.

Protein	Uniprot Accession Number	TP3 Protein Result as % of TP2 Baseline; Median (Range)	Number of Patients with Decrease	Number of Patients with Decrease > IQR of Baseline Variability	*p* Value	*q* Value (FDR)
TNF-related apoptosis inducing ligand (TRAIL)	P50591 *	72.3 (45.1–132.5)	23 (95.8%)	22 (91.7%)	<0.001	<0.001
C-X-C motif chemokine 10 (CXCL10)	P02778	64.1 (42.3–186)	21 (87.5%)	20 (83.3%)	<0.001	<0.001
C-X-C motif chemokine 11 (CXCL11)	O14625	55.8 (2.8–155)	23 (95.8%)	20 (83.3%)	<0.001	<0.001
Interferon gamma (IFN-gamma)	P01579	55.0 (8.8–102.2)	20 (83.3%)	20 (83.3%)	<0.001	<0.001
Mothers against decapentaplegic homolog 5 (MAD homolog 5)	Q99717	83.0 (62.9–126.6)	21 (87.5%)	18 (75%)	<0.001	<0.001
Monocyte chemotactic protein 2 (MCP-2)	P80075	72.3 (45.1–146.4)	22 (91.7%)	17 (70.8%)	<0.001	<0.001
Monocyte chemotactic protein 4 (MCP-4)	Q99616	59.2 (13.5–146.5)	23 (95.8%)	17 (70.8%)	<0.001	<0.001
Fas ligand (FASL)	P48023 *	80.2 (66–116.8)	20 (83.3%)	16 (66.7%)	<0.001	0.001
Matrix metalloproteinase-12 (MMP-12)	P39900	77.9 (52.4–124.7)	20 (83.3%)	16 (66.7%)	<0.001	<0.001
Proto-oncogene tyrosine-protein kinase receptor Ret (RET)	P07949	79.2 (50.5–213.4)	20 (83.3%)	15 (62.5%)	0.002	0.007
Cathepsin L2 (CTSV)	O60911	77.5 (11.9–117.3)	18 (75%)	15 (62.5%)	<0.001	0.004
Vascular endothelial growth factor receptor 2 (VEGFR-2)	P35968	91.4 (72.7–127.9)	20 (83.3%)	14 (58.3%)	0.001	0.003
Angiopoietin-1 (ANGPT-1)	Q15389	63.4 (23–199.7)	19 (79.2%)	14 (58.3%)	0.003	0.005
Platelet-derived growth factor subunit B (PDGF subunit B)	P01127	69.3 (32–208)	20 (83.3%)	12 (50%)	<0.001	0.002
Natural killer cell receptor 2B4 (CD244)	Q9BZW8	84.4 (64.6–181.3)	17 (70.8%)	11 (45.8%)	0.004	0.007
Fibroblast growth factor 2 (FGF-2)	P09038	81.4 (1.8–200.2)	18 (75%)	11 (45.8%)	0.001	0.003
C-X-C motif chemokine 5 (CXCL5)	P42830	61.0 (15.7–358.5)	20 (83.3%)	9 (37.5%)	<0.001	0.002

## Data Availability

The data presented in this study are available on request from the corresponding author subject to ethical restrictions of any patient.

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
