# Peer review of "An Exploratory Analysis of Changes in Circulating Plasma Protein Profiles Following Image-Guided Ablation of Renal Tumours Provides Evidence for Effects on Multiple Biological Processes"

_cancers, 2021, doi:10.3390/cancers13236037_

Round 1

Reviewer 1 Report

The work is very interesting and emerging. Further studies are required to confirm those results because the sample size is quite small, but adequate in relation to the fact that the vast majority of small renal masses are surgically treated.

Author Response

Response 1. We thank the reviewer for this review

Reviewer 2 Report

The Authors of cancers-1417815 manuscript entitled “An exploratory analysis of changes in circulating plasma protein profiles following image-guided ablation of renal tumours provides evidence for effects on multiple biological processes” performed interesting proteomic experiment in patients with renal tumours subjected to treatment with image–guided ablation with heat-based or cold-based techniques. The whole conception is consistent and was justified by the results of previous model studies suggesting immune and inflammatory responses of the organism generated by the ablation procedure. The Authors used a systems approach, with different time points, allowing to investigate the time-depended changes in proteome profiling. Analytical data preparation included normalization to take into account inter-individual and intra-individual variability. The Authors placed their research in a broader context and provided strengthens and weaknesses of their study as well as some possibilities of the continuation or practical application of their study. There are only some minor inconsistencies and ambiguities, which should be addressed before publication. They have been listed below. I recommend minor revision of this manuscript.

  • In my opinion Figures 2 and 3 should be included in supplementary materials
  • Data presenting at Figure 5 (scatterplots) concerning the dependences of the number of cryoprobes with the levels of some investigated proteins are unclear for me. Especially taking into account their practical usefulness. The Authors should explain it more thoroughly.
  • It is well known phenomenon that existing tumour changes the environment of surrounding tissues and exerts the systemic influence on the whole organism. The Authors focused on the targeted proteome analyses of proteins related with immune system response and inflammatory process. However, renal tumours and performed treatments may also influence lipidome profile, which in turn also interact with immune system and inflammatory process. This should be discussed in ‘discussion’ section especially as one of the possible ways of further studies.

Author Response

Response to Reviewer 2 Comments

The Authors of cancers-1417815 manuscript entitled “An exploratory analysis of changes in circulating plasma protein profiles following image-guided ablation of renal tumours provides evidence for effects on multiple biological processes” performed interesting proteomic experiment in patients with renal tumours subjected to treatment with image–guided ablation with heat-based or cold-based techniques. The whole conception is consistent and was justified by the results of previous model studies suggesting immune and inflammatory responses of the organism generated by the ablation procedure. The Authors used a systems approach, with different time points, allowing to investigate the time-depended changes in proteome profiling. Analytical data preparation included normalization to take into account inter-individual and intra-individual variability. The Authors placed their research in a broader context and provided strengthens and weaknesses of their study as well as some possibilities of the continuation or practical application of their study. There are only some minor inconsistencies and ambiguities, which should be addressed before publication. They have been listed below. I recommend minor revision of this manuscript.

Response: Thank you to the reviewer for their helpful and positive review

Point 1: In my opinion Figures 2 and 3 should be included in supplementary materials

Response 1:  The full profiles of all the proteins are already included in Supplementary Materials. The data in Figures 2 and 3 are selected examples from this and included in the main body of the paper so that the reader can easily see and understand key points of the results.  We feel therefore that these 2 figures are central to the message of the paper and would respectfully disagree with the reviewer.

Point 2:  Data presenting at Figure 5 (scatterplots) concerning the dependences of the number of cryoprobes with the levels of some investigated proteins are unclear for me. Especially taking into account their practical usefulness. The Authors should explain it more thoroughly.

Response 2:  We have now included more explanation regarding this data in the Results and Discussion sections.  Essentially we think these correlations illustrate the greater inflammatory response from the larger ablation zone generated with increasing number of cryoprobes. From a clinical perspective, this may be helpful in terms of these proteins providing a readout which may help in the standardisation of treatment approaches and/or assessment of the magnitude and duration of the biological response of the ablative element which will be important in optimising combination therapy strategies.

Point 3:  It is well known phenomenon that existing tumour changes the environment of surrounding tissues and exerts the systemic influence on the whole organism. The Authors focused on the targeted proteome analyses of proteins related with immune system response and inflammatory process. However, renal tumours and performed treatments may also influence lipidome profile, which in turn also interact with immune system and inflammatory process. This should be discussed in ‘discussion’ section especially as one of the possible ways of further studies.

Response 3: We completely agree with this point and have now included some referenced discussion of this point towards the end of the Discussion section.

Reviewer 3 Report

Rev:

The current original work describes the effect of a panel of ablation techniques on circulating plasma proteome in patients with renal cancer. While the topic is important in the field of cancer biomarkers, the authors should revise their manuscript in order to highlight information obtained in a clinical setting in contrast to data obtained in a pure in vitro setting. Furthermore, this exploratory study shows as a fishing expedition without a clear and strong hypothesis. The following points need to be addressed:

  • Results and discussion related to each increased or decreased analyte are pretty speculative, with no solid science background.
  • While “intraindividual variability” may be a reason for presenting the data as percentage increment/ reduction, the actual real protein concentration for any analyte investigated must be showed (as supplementary) with the stats. Moreover “Protein concentration as % of baseline” definition in the head of the tables 2 and 3 does not make any sense. A concentration cannot be a percentage.
  • The study also miss a very important control group, untreated patients. I would include as reference the concentration range of these proteins from a group a healthy volunteers.
  • In the methods section is mentioned that serum and plasma were collected. What was the reason for serum collection? There are no data related to sera investigation. Why plasma was preferred over serum for the proteomic investigation?
  • There is no information about co-morbidities differences between the investigated groups (especially immune related disease).
  • There is no information about the possible side effects occurred post-treatment and a discussion about their effect on proteomics (if). For example dehydration and fever are known for changing proteome profiles in plasma.
  • There is no discussion about the potential contribution of anesthesia and the post-ablation antibiotic in the observed results, when compare the groups.
  • Are the observed changes in plasma proteome a direct consequence of ablation method on the tumor microenvironment or an indirect consequence via suppressing the cancer which dampen normal immunity? Are the observed increment/reduction observed a consequence of the ablation treatment or just a biological random effect? These should be discussed.
  • Finally, their observation should be integrated in the natural history of renal cancer and its interaction with immune system.

Author Response

Response to Reviewer 3 Comments

The current original work describes the effect of a panel of ablation techniques on circulating plasma proteome in patients with renal cancer. The following points need to be addressed:

Point 1:  While the topic is important in the field of cancer biomarkers, the authors should revise their manuscript in order to highlight information obtained in a clinical setting in contrast to data obtained in a pure in vitro setting. Furthermore, this exploratory study shows as a fishing expedition without a clear and strong hypothesis.

Response 1:  We apologise if the clinical setting of the study and the interpretation of the results in that context was inadequate. We have now included more clinical information, some also in response to specific points below, and addressed the data more in the context of the clinical setting.  This is very much an exploratory study as we have described and inevitably with the approach taken (as with any exploratory ‘omics study) there is an element of “fishing”. This is a strength of such studies in terms of enabling new knowledge but agree that such studies must have an underlying hypothesis and appropriate design and we have now clearly stated the underlying hypothesis more clearly.

Point 2:  Results and discussion related to each increased or decreased analyte are pretty speculative, with no solid science background.

Response 2:  We agree that there is some speculation with regard to the changes in analytes which is inevitable with novel findings but we have tried to back this off known scientific knowledge where possible and have extended this in some areas.

Point 3:  While “intraindividual variability” may be a reason for presenting the data as percentage increment/ reduction, the actual real protein concentration for any analyte investigated must be showed (as supplementary) with the stats. Moreover “Protein concentration as % of baseline” definition in the head of the tables 2 and 3 does not make any sense. A concentration cannot be a percentage.

Response 3:  A strength of this study and one which is lacking in many studies is the use of two baseline samples prior to treatment so that the level of intraindividual variability of the different proteins can be appreciated and allow interpretation of any potential intervention-induced changes against that background. Given the variability of baseline concentrations expressing the data as %change allows intervention-induced changes to be more easily visualised.  However, we agree with the reviewer that showing the concentration plots for each protein across the timepoints i.e. not expressed as a percentage of the baseline, also provides useful information and we have now included additional supplementary data files to show this in terms of both profiles and stats. We agree that the heading in Tables 2 and 3 is not clear in terms of our relative expression of concentrations and have addressed this.

Point 4:  The study also miss a very important control group, untreated patients. I would include as reference the concentration range of these proteins from a group a healthy volunteers.

Response 4:  Although each patient’s own baseline values are the most important reference point to allow determination of the impact of the ablation, we agree that knowing what the protein concentrations are in the patients compared with healthy controls provides valuable additional information. In the new supplementary figures provided above in point 3 (Figures S1 and 2), we have also included protein data from a group (n=11) of healthy controls of similar age range and whose samples were processed according to the same SOP and analysed at the same time so that direct comparisons can be made. These are obviously inadequate to provide a full reference range which would require many more samples but will provide an indication of whether the patients have similar levels of the proteins at baseline to healthy controls or are elevated already either as a consequence of their cancer or comorbidities. We have altered the methods section of the manuscript to reflect this, also included a supplementary table (Table S2) of the baseline results in the RCC patients compared with healthy controls and discussed this in the paper.

Point 5:  In the methods section is mentioned that serum and plasma were collected. What was the reason for serum collection? There are no data related to sera investigation. Why plasma was preferred over serum for the proteomic investigation?

Response 5:  This has now been clarified in the Methods section. We usually standardly obtain both serum and plasma from patients in most of our research studies as they are useful for different analyses. As we used plasma in this study to minimise the impact of platelet activation/lysis (which occurs with serum preparation) on some of the proteins measured we have just described that in the Methods now to avoid confusion.

Point 6:  There is no information about co-morbidities differences between the investigated groups (especially immune related disease).

Response 6:  We have now included more clinical details in a Supplementary table (Table S1) and referred to this in the text so that the absence of any marked differences between groups in terms of this aspect can be appreciated. We have also generated new Supplementary longitudinal profile figures highlight those patients in particular in the profiles shown so that any potential impact of the complications can be appreciated easily. We have also included more discussion about co-morbidities and treatment impacts and data interpretation in the paper.

Point 7:  There is no information about the possible side effects occurred post-treatment and a discussion about their effect on proteomics (if). For example dehydration and fever are known for changing proteome profiles in plasma.

Response 7:  We have now included information in the above supplementary data file (Table S1) for 3 patients who experienced complications post-ablation and have generated additional supplementary figures (Figures S8-S11) highlighting the profiles for those 3 patients for each protein so the potential impact can be seen. We have also now discussed the potential impact of such complications on the results seen and the importance of such factors in the interpretation of the results.

Point 8:  There is no discussion about the potential contribution of anesthesia and the post-ablation antibiotic in the observed results, when compare the groups.

Response 8:  All patients underwent general anaesthesia and the same protocol-driven antibiotic treatment as already described in the Methods and so the potential impact of these should be similar across patients, although one patient with fever post-ablation received additional antibiotic treatment. The impact of anaesthesia generally on circulating proteins is poorly understood as most studies investigating interventions obtain samples at a time which is actually post-surgery in addition to anaesthesia.  However, we have now included some discussion of this point and presented some evidence from published studies which support a likely minimal impact of anaesthesia per se on most proteins analysed. 

Point 9:  Are the observed changes in plasma proteome a direct consequence of ablation method on the tumor microenvironment or an indirect consequence via suppressing the cancer which dampen normal immunity? Are the observed increment/reduction observed a consequence of the ablation treatment or just a biological random effect? These should be discussed. Finally, their observation should be integrated in the natural history of renal cancer and its interaction with immune system.

Response 9:  We have included more discussion on these points.

Round 2

Reviewer 3 Report

I would like to thank the authors for the implementation of changes suggested by the reviewer. The revised manuscript is responsive to reviewer’ comments including adding new important information and comprehensive integration of these new data. I believe that this updated manuscript is suitable for publication now.